# Representative Demonstration Selection for In-Context Learning with Two-Stage Determinantal Point Process

**Zhao Yang[1,2], Yuanzhe Zhang[1,2], Dianbo Sui[4], Cao Liu[3], Jun Zhao[1,2], Kang Liu[1,2,5]**

[1]School of Artificial Intelligence, University of Chinese Academy of Sciences
[2]The Laboratory of Cognition and Decision Intelligence for Complex Systems,
Institute of Automation, Chinese Academy of Sciences,[3]Meituan, Beijing
[4]Harbin Institute of Technology, Weihai, [5]Shanghai Artificial Intelligence Laboratory
{zhao.yang, yzzhang, jzhao, kliu}@nlpr.ia.ac.cn
suidianbo@hit.edu.cn, liucao@meituan.com

## Abstract

Although In-Context Learning has proven effective across a broad array of tasks, its efficiency is noticeably influenced by the selection of demonstrations. Existing methods tend to select different demonstrations for each test instance, which is time-consuming and poses limitations in practical scenarios. Therefore, this study aims to address the challenge of selecting a representative subset of in-context demonstrations that can effectively prompt different test instances in a specific task. We propose that this representative subset should be of high quality and diversity. Our empirical analyses confirm that demonstrations that meet these criteria can indeed bolster model performance. To satisfy these criteria, this paper further introduces a two-stage Determinantal Point Process (DPP) method designed to incorporate both quality and diversity in the process of demonstration selection, thereby obtaining representative in-context demonstrations. Through comprehensive experimentation, we have confirmed the efficacy of our proposed method, paving the way for more practical and effective In-Context Learning.

## 1 Introduction

The emergence of Large Language Models (LLMs) has swept various Natural Language Processing (NLP) tasks and brought new research paradigms (Bommasani et al., 2021; Wei et al., 2022; OpenAI, 2023). In-Context Learning (ICL) (Brown et al., 2020) is the primary paradigm for leveraging LLMs, which enables the model to generalize rapidly from a few examples without parameter update. Recent studies have shown that ICL has significantly improved over the zero/few-shot setting and even surpassed the performance of full data fine-tuning (Dong et al., 2022).

However, ICL is insufficiently stable and robust in performance when compared to traditional fine-tuning, and exhibits sensitivity to numerous factors, including demonstration selection (Liu et al.,

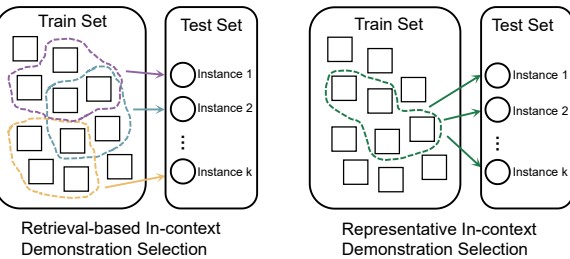

Figure 1: Comparison between existing retrieval-based in-context demonstration selection and our representative in-context demonstration selection.

2022; Rubin et al., 2022), demonstration formats (Min et al., 2022), demonstration labels (Zhao et al., 2021; Yoo et al., 2022), and demonstration order (Lu et al., 2022). Among these factors, demonstration selection is the most significant, as they provide only information about the task (Liu et al., 2022). Therefore, selecting suitable demonstrations is necessary.

Existing demonstration selection methods usually adopt retrieval-based solutions, which aim to select different demonstrations for each test instance (Liu et al., 2022; Sorensen et al., 2022; Rubin et al., 2022; Zhang et al., 2022a). However, these retrieval-based methods can be costly in practice. In detail, these methods can only perform inference on one sample at a time, which incurs high costs in terms of token usage, particularly when considering leveraging batch prompting (Cheng et al., 2023). Besides, performing demonstration selection for each testing example is obviously time-costing (§4.3). Based on this, as shown in Figure 1, this paper aims to select a representative demonstration subset from a pool of training examples for all test instances in a specific task rather than each test instance.

To select a representative demonstration subset for a task, we argue that there are two criteria that should be satisfied: *quality* and *diversity*. Firstly, in ICL, the *quality* of an instance is defined as the

degree that it can help the LLM to make correct predictions. Intuitively, high-*quality* instances can improve the LLM's performance on as many test examples as possible. Secondly, *diversity* means that the elements in the selected demonstration subset should be mutually dissimilar and represent the training set as comprehensively as possible. Moreover, in this paper, we argue that *diversity* can be refined into *semantic diversity* and *influence diversity*. *Semantic diversity* means that the selected examples should be diverse enough to cover more semantics or expressions in the given training set, which is consistent with existing representative subset selection in traditional deep learning (Coleman et al., 2020; Guo et al., 2022). *Influence diversity* means that each selected example should help the model correctly classify a diverse set of testing samples, enabling the total demonstration set to support the correct classification of more test instances for the given LLM. Pilot experiments (§2) also verify that the demonstration subsets satisfying these criteria could improve ICL performance.

In this paper, we make the following efforts to meet the above criteria. To achieve the *quality* criterion, we resort to recent instance-based explanation methods (Koh and Liang, 2017; Pruthi et al., 2020). Different from them, this paper extends existing instance-based explanation methods to ICL and defines the influence score to measure the contribution of a training example to the correct classification of another example. To achieve the *diversity* criterion, this paper seeks the help of the determinantal point process (DPP), which is a probabilistic model that could maintain high diversity among different instances by pairwise repulsion between instances (Kulesza et al., 2012).

Moreover, to incorporate both high *quality* and *diversity* for demonstration selection, this paper further proposes a two-stage DPP selection method. In specific, in the first stage (§3.2.1), we utilize the semantic information measure instance similarity and then leverage DPP to obtain a subset of instances with high *semantic diversity*. In the second stage (§3.2.2), we first compute the *quality* score of each instance in this subset based on our defined influence scores (§2.2), which could reduce a significant amount of meaningless inference time compared to computing the quality for each training example. Then we utilize the obtained influence scores to measure influence similarity between instances and construct another DPP. Besides, we also introduce

the quality scores (§2.2) of these examples to this DPP. As a result, both quality and influence diversity could be optimally introduced in the selection process. This two-stage DPP selection method enables the selected demonstration subset could not only cover more semantics in the training set but also help the given LLM classify more examples with better performance.

Our contributions can be summarized as follows:

- This paper aims to select representative in-context demonstrations and propose two criteria (*quality* and *diversity*) that such demonstrations should satisfy.

- To satisfy the proposed criteria, this paper proposes a two-stage DPP selection method. According to our method, the selected demonstrations could not only be representative of the large training set but also facilitate the model's generalization to new instances.

- Extensive experimental results demonstrate the effectiveness of our proposed method. And we also show the significant advantage in inference time and token usage compared to retrieval-based methods.

## 2 Pilot Experiments

This section explores whether the proposed criteria could improve the performance of ICL. In specific, we measure the impact of three factors: semantic diversity, instance quality, and influence diversity. We choose SST-2 (Socher et al., 2013) as the test dataset and conduct 4-shot experiments on GPT2-xl (1.5B) (Radford et al., 2019), GPT-J (6B) (Wang, 2021) and GPT-NeoX (20B) (Black et al., 2022).

### 2.1 Impact of Semantic Diversity

To explore the impact of semantic diversity, we firstly utilize sentence-BERT (Reimers and Gurevych, 2019) to encode all train instances. Then, with these sentence representations, we use K-means (Hartigan and Wong, 1979) to partition all train instances into 4 clusters. Finally, we design three strategies with different semantic diversity: 1) `single-cluster`: we sample 4 examples from each cluster. 2) `random`: random sampling 4 examples from the whole training set. 3) `multi-cluster`: we sample one example from each of these 4 clusters. Obviously, `multi-cluster` has the best semantic diversity and `single-cluster` has the worst diversity.

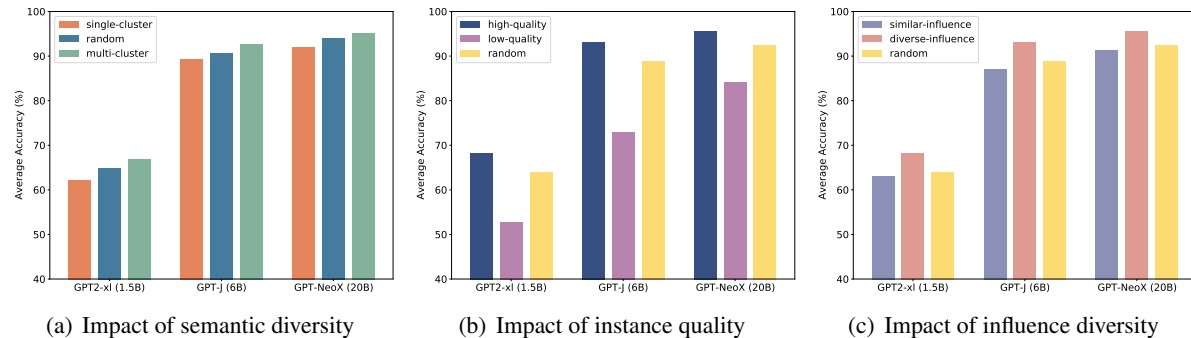

(a) Impact of semantic diversity     (b) Impact of instance quality     (c) Impact of influence diversity

Figure 2: Comparisons of the three factors: semantic diversity, instance quality, and influence diversity. We list the 4-shot performance on SST-2 for three LLMs: GPT2-xl (1.5B), GPT-J (6B), and GPT-NeoX (20B).

Figure 2(a) presents the corresponding results of the five sample strategies. From the figure, we observe that `multi-cluster` is always superior to the other two strategies across these three LLMs. According to these results, we could find that better semantic diversity could lead to higher accuracy across models of different scales.

## 2.2 Impact of Instance Quality

To measure the quality of an instance, we need to define the influence score of an instance at first. Existing example-based explanation methods (Koh and Liang, 2017; Pruthi et al., 2020) define the influence of an example in a training set as the prediction difference of the test set between the model trained with and without this example. Inspired by this, we define the influence of the demonstration $e$ on another example $e_i = (x_i, y_i)$ in ICL as follows:

$$\mathrm{Inf}(e, e_i) = p_{\mathrm{LM}}(y_i|e, x_i) - p_{\mathrm{LM}}(y_i|x_i) \quad (1)$$

where $y_i$ is the golden label of example $x_i$. $p_{\mathrm{LM}}(y_i|e, x_i)$ and $p_{\mathrm{LM}}(y_i|x_i)$ refers to the output probability of $y_i$ conditioned on $(e, x)$ and $x$ for a given language model LM, respectively.

$\mathrm{Inf}(e, e_i)$ could measure the contribution of $e$ to the correctness of the example $e_i$. Based on this, we define the quality metric of the demonstration $e$ as follows:

$$\mathrm{Q}(e) = \frac{\sum_{e_i \in \mathcal{D}_{\mathrm{score}}} \mathrm{Inf}(e, e_i)}{T} \quad (2)$$

where $\mathcal{D}_{\mathrm{score}}$ denotes the score set used for measuring quality and T refers to its size. $\mathrm{Q}(e)$ characterizes the contribution of $e$ to the model performance on the score set. The higher the value of this metric, the more $e$ can help the language model in making correct predictions on this score set.

To improve efficiency, we randomly choose 100 examples and measure their quality by randomly sampling an additional 200 samples as the score set. Then we design three strategies with different quality: 1) `high-quality`: we choose the top 4 examples with the highest quality as demonstrations. 2) `low-quality`: we choose the bottom 4 examples with the lowest quality as demonstrations. 3) `random`: we randomly choose 4 examples from these 100 examples as demonstrations.

Figure 2(b) shows the experimental results. From these results, we find that demonstrations with higher quality could get better performance.

## 2.3 Impact of Influence Diversity

With the same setting in §2.2, we have obtained the influence scores on 200 score instances, which could be seen as a 200-dimension vector. Based on this, we can get the influence vector for each instance and then we utilize the similar cluster method in §2.1 to partition the 100 examples into 4 clusters. We construct the following sampling strategies with different influence diversity: 1) `similar-influence`: we sample 4 examples from each cluster and average their performance. 2) `diverse-influence`: we sample one example from each cluster as demonstrations. 3) `random` randomly sample 4 examples as demonstrations.

Figure 2(c) shows the corresponding results. According to these results, we observe that demonstrations with better influence diversity could improve ICL performance for LLMs of different scales.

## 2.4 The Conclusion on Pilot Experiments

Our three experiments show that *semantic diversity*, *instance quality*, and *influence diversity* all can improve ICL performance, which verifies the effectiveness of the proposed criteria. Therefore, our

method would try to introduce these three factors in the selection process.

# 3 Method

In §3.1, we illustrate the formulation of in-context learning and the background of the determinantal point process. Then, we introduce the implementation of our method in §3.2, which could introduce the three factors in the whole selection process.

## 3.1 Preliminary

### 3.1.1 In-context Learning

In-context learning is the core emergent ability of LLMs and only requires a few examples and corresponding labels to solve a task. Formally, a demonstration $e_i = (x_i, y_i)$ consists of an instance $x_i$ and its label $y_i$. For a new test instance $x_{test}$, $K$-shot in-context learning generate its label $y_{predict}$ as follows:

$$P_{\text{LM}}(y_{predict}|e_1, e_2, \cdots, e_K, x_{test}) \qquad (3)$$

where $e_i$ is sampled from the whole train dataset and $\{e_1, e_2, \cdots, e_K\}$ is usually called context.

### 3.1.2 Determinantal Point Process

Determinantal Point Process (DPP) is an elegant probabilistic model with the ability to express negative interactions (Kulesza et al., 2012). DPP could help us find a representative subset while keeping high diversity among different items (Chen et al., 2018). Formally, for an index set $A = \{1, 2, \cdots, M\}$ and its corresponding item set $I_A = \{a_1, a_2, \cdots, a_M\}$, DPP provides a probability measurement for $2^N$ item subsets. Given the feature representation $\boldsymbol{a}_i$ for the item $a_i$, DPP computes a positive semi-definite (PSD) matrix $\boldsymbol{L} \in \mathbb{R}^{M \times M}$, where $\boldsymbol{L}_{ij} = k(\boldsymbol{a}_i, \boldsymbol{a}_j)$ for an kernel function $k(\cdot, \cdot)$ and $\boldsymbol{L}_{ij}$ usually measures the correlation between $a_i$ and $a_j$. Based on this, if a subset is determined by the index set $Y \subseteq A$, the probability of selecting Y could be defined as follows:

$$P(Y) = \frac{\det(\boldsymbol{L}_Y)}{\det(\boldsymbol{L} + \boldsymbol{I})} \qquad (4)$$

where $\boldsymbol{L}_Y$ refers to a submatrix of $\mathbf{L}$ and consists of $\boldsymbol{L}_{ij}$ for $i, j \in Y$. $\mathbf{I}$ is an identity matrix and $\det(\cdot)$ is the determinant of a matrix.

Under this distribution, we could select the representative subset $Y_{\text{best}}$ of size $k$ as follows:

$$Y_{\text{best}} = \underset{Y \subseteq A, |Y|=k}{\operatorname{argmax}} \det(\boldsymbol{L}_Y) \qquad (5)$$

Obviously, the maximum a posteriori inference in DPP has high time complexity (Kulesza and Taskar, 2011). Following Chen et al. (2018), the time complexity of selecting a subset of size k is $O(k^2 M)$.

The core of DPP is how to obtain the corresponding PSD matrix. Existing studies usually incorporate task-relevant information into this matrix, in order to utilize DPP to achieve diversity.

## 3.2 Two-Stage DPP Selection Method

In this section, we introduce a novel two-stage DPP selection method, which could introduce semantic diversity, high quality, and influence diversity in demonstration selection.

### 3.2.1 First Stage: Selecting Candidate Subset with Semantic Diversity

According to Equation 1, computing the influence score requires inference time. Thus, computing the influence score for each instance in the training set is infeasible for its high costs. Intuitively, we need to reduce the size of the set that requires calculating influence scores.

Based on this, this paper selects a small subset from the original training set by introducing semantic diversity. Formally, the training set $\mathcal{D} = \{e_1, e_2, \cdots, e_N\}$ contains $N$ instances where $e_i = (x_i, y_i)$. We firstly utilize sentence-BERT (Reimers and Gurevych, 2019) to encode these instances and obtain their semantic representations $\{\boldsymbol{x}_1, \boldsymbol{x}_2, \cdots, \boldsymbol{x}_N\}$. Then we could easily get the dataset representation $\boldsymbol{s} \in \mathbb{R}^{N \times d}$ by stack, where $d$ is the dimension of each sentence. Finally, we could get the PSD matrix $\boldsymbol{L}_S$ in DPP as follows:

$$\boldsymbol{L}_S = \boldsymbol{s}\boldsymbol{s}^T \qquad (6)$$

Obviously, $\boldsymbol{L}_S \in \mathbb{R}^{N \times N}$ is a real symmetric matrix. With $\boldsymbol{L}_S$, we could obtain a candidate subset $\mathcal{D}_{\text{sem}}$ of size $N_{\text{sem}}$ according to Equation 5.

### 3.2.2 Second Stage: Selecting Demonstrations via High Quality and Influence Diversity

In previous parts, we have introduced semantic diversity into $\mathcal{D}_{\text{sem}}$. In the following parts, we would illustrate how to select high-quality instances while maintaining influence diversity to form the final demonstration set.

Firstly, we need to select the score set to measure the quality of an instance in $\mathcal{D}_{\text{sem}}$ according to Equation 2. Recent probing studies (Min et al., 2022; Yoo et al., 2022) show the importance of the label for ICL. Therefore, we only require the

**Algorithm 1** Two-Stage DPP Selection

**Input:** Training set $\mathcal{D} = \{e_i\}_{i=1}^N$, language model LM, candidate subset size $N_{\text{sem}}$, score set size $T$, demonstration set size $k$, sentence encoder sbert.
**Output:** Demonstration Subset $\mathcal{D}_{\text{dem}}$
1: **for** $e_i \in \mathcal{D}$ **do**
2: $\quad \boldsymbol{x}_i = \text{sbert}(x_i)$
3: **end for**
4: $\boldsymbol{s} = \text{stack}(\boldsymbol{x}_1, \boldsymbol{x}_2, \cdots, \boldsymbol{x}_N)$
5: $\boldsymbol{L}_S = \boldsymbol{s}\boldsymbol{s}^T$
6: $\mathcal{D}_{\text{sem}} = \arg\max_{Y \subseteq \mathcal{D}, |Y| = N_{\text{sem}}} \frac{\det(\boldsymbol{L}_Y)}{\det(\boldsymbol{L}_S + \boldsymbol{I})}$
7: random sampling $\mathcal{D}_{\text{score}}$ of size $T$ from $\mathcal{D}/\mathcal{D}_{\text{sem}}$
8: **for** $e_i \in \mathcal{D}_{\text{sem}}$ **do**
9: $\quad$ **for** $e_j \in \mathcal{D}_{\text{score}}$ **do**
10: $\qquad \text{Inf}(e_i, e_j) = p_{\text{LM}}(y_j | e_i, x_j) - p_{\text{LM}}(y_j | x_j)$
11: $\quad$ **end for**
12: $\quad \text{Q}(e_i) = \frac{\sum_{e_j \in \mathcal{D}_{\text{score}}} \textbf{Inf}(e_i, e_j)}{T}$
13: $\quad \boldsymbol{e_i} = (\text{Inf}(e_i, e_1), \text{Inf}(e_i, e_2), \cdots, \text{Inf}(e_i, e_T))$
14: **end for**
15: $\boldsymbol{I} = \text{stack}(\boldsymbol{e}_1, \boldsymbol{e}_2, \cdots, \boldsymbol{e}_{N_{\text{sem}}})$
16: $\boldsymbol{Q} = (\text{Q}_{e_1}, \text{Q}_{e_2}, \cdots, \text{Q}_{e_{N_{\text{sem}}}})$
17: $\boldsymbol{L}_I = \boldsymbol{Q} \cdot \boldsymbol{I}\boldsymbol{I}^T \cdot \boldsymbol{Q}$
18: $\mathcal{D}_{\text{dem}} = \arg\max_{Y \subseteq \mathcal{D}_{\text{sem}}, |Y| = k} \frac{\det(\boldsymbol{L}_Y)}{\det(\boldsymbol{L}_I + \boldsymbol{I})}$
19: **return** $\mathcal{D}_{\text{dem}}$

---

examples in the scoring set to include all of the labels. With this premise satisfied, we obtained the score set $\mathcal{D}_{\text{score}}$ of size $T$ through a random sample from $\mathcal{D}/\mathcal{D}_{\text{sem}}$. With the score set, we could easily compute the influence score and quality of each instance in $\mathcal{D}_{\text{sem}}$ according to Equation 1 and 2.

For item $x_j \in \mathcal{D}_{\text{sem}}$, we could obtain $T$-dimension influence embedding $\boldsymbol{I_j}$ where each dimension denotes its influence on each instance in the score set and its quality score $\text{Q}(e_j)$. Then, we can obtain the total influence representation matrix $\boldsymbol{I} \in \mathbb{R}^{N_{\text{sem}} \times T}$ and the quality representation of $\boldsymbol{Q} \in \mathbb{R}^T$ by stacking and contacting operation. Thus we can compute another PSD matrix $\boldsymbol{L}_I$ as follows:

$$\boldsymbol{L}_I = \boldsymbol{Q} \cdot \boldsymbol{I}\boldsymbol{I}^T \cdot \boldsymbol{Q} \qquad (7)$$

Actually, $\boldsymbol{I}\boldsymbol{I}^T$ could construct a basic PSD matrix to help us introduce influence diversity. Inspired by Chen et al. (2018), we further introduce $\boldsymbol{Q}$ to help us select high-quality instances. In this way, DPP which is based on $\boldsymbol{L}_I$ could help us select high-quality and diverse examples. Concretely, for $k$-shot ICL, we can select a demonstration subset $\mathcal{D}_{\text{dem}}$ of size $k$ according to Equation 5.

In summary, Algorithm 1 presents the whole process of our two-stage DPP selection method. Figure 9 shows a more direct illustration of our method. Besides, a recent study (Lu et al., 2022) shows the ordering of demonstrations also has a significant influence on the performance. But in

this paper, we only focus on demonstration selection and just determine the order in ascending order of their quality scores in $\mathcal{D}_{\text{score}}$.

## 4 Experiments

### 4.1 Experimental Setups

#### 4.1.1 Datasets

Following previous studies (Zhao et al., 2021; Liu et al., 2022), we conduct experiments on six text classification tasks: SST-2 (Socher et al., 2013), TREC (Voorhees and Tice, 2000), CB (De Marneffe et al., 2019), AGNews (Zhang et al., 2015), DBPedia (Zhang et al., 2015) and RTE (Dagan et al., 2006). And we utilize accuracy as the evaluation metric. Statistics and the prompt format for each dataset could be found in Appendix A.

#### 4.1.2 Large Language Models

This paper conducts experiments on three large language models of different scales: GPT2-xl (1.5B) (Radford et al., 2019), GPT-J (6B) (Wang, 2021) and GPT-NeoX (20B) (Black et al., 2022).

#### 4.1.3 Baselines

In this paper, we compare our method with the following baseline methods: **Representative Subset Selection in Deep Learning**: Least Confidence (LC) (Coleman et al., 2020) selects representative examples according to the max probability on all labels. Cal (Margatina et al., 2021) selects representative examples with the biggest logits divergence from neighbors. **Recent Representative Demonstration Selection Methods**: Random selects demonstrations by randomly sampling. Cluster (Zhang et al., 2022b; Gao et al., 2023) selects $k$ demonstrations by choose $k$ cluster centers of K-means cluster by semantic. DPP (Levy et al., 2023) utilize the semantic information select k demonstrations according to k-DPP (Kulesza and Taskar, 2011; Chen et al., 2018).

### 4.2 Main Results

Table 1 presents the corresponding results. Traditional representative subset selection methods get poor performance in ICL settings. Least Confidence even gets a worse performance than random selection and Cal just gets a comparable performance with random selection. Our designed baselines Cluster and DPP, which both introduce semantic diversity, achieve some improvements compared to random selection. And DPP could

| Model | Method | SST-2 | Trec | CB | AGNews | DBpedia | RTE | AVG |
|---|---|---|---|---|---|---|---|---|
| | | 2/4/8 | 2/4/8 | 2/4/8 | 2/4/8 | 2/4/8 | 2/4/8 | 2/4/8 |
| GPT2-xl 1.5B | LC | 52.14/ 53.72/55.86 | 18.20/22.40/36.20 | 43.78/44.85/46.75 | 37.62/38.48/41.27 | 39.45/46.74/52.08 | 48.95/51.02/50.24 | 40.02/42.86/47.06 |
| | Cal | 53.88/56.04/59.17 | 19.10/24.30//38.10 | 45.86/46.55/46.62 | 41.02/42.58/45.12 | 54.36/61.32/69.56 | 50.22/49.35/50.89 | 44.07/46.70/51.58 |
| | Random | 52.33/55.27/56.97 | 17.36/23.76/37.20 | 46.43/46.47/46.57 | 40.50/41.86/43.84 | 55.80/61.24/68.06 | 51.84/52.20/52.56 | 44.04/46.80/50.87 |
| | Cluster | 53.08/55.43/60.37 | 18.34/26.40/44.97 | 47.38/48.76/52.15 | 63.57/64.89/68.02 | 59.33/67.28/75.47 | 51.33/50.96/51.76 | 48.84/50.29/58.79 |
| | DPP | 53.02/56.12/63.59 | 18.20/27.60/50.80 | 47.07/50.00/55.36 | 63.88/66.72/69.86 | 58.48/69.44/78.77 | 51.57/51.01/53.43 | 48.70/53.48/61.97 |
| | Ours | 59.86/64.31/72.40 | 32.68/38.60/57.22 | 48.21/56.07/59.14 | 67.21/73.91/76.42 | 65.93/67.74/75.35 | 53.07/53.79/53.77 | 54.49/59.07/65.72 |
| GPT-J 6B | LC | 81.87/84.76/88.02 | 34.20/37.10/40.60 | 24.45/50.27/53.87 | 53.87/54.98/66.04 | 60.96/76.65/80.47 | 50.07/51.88/52.24 | 50.90/59.27/63.54 |
| | Cal | 84.35/89.04/89.57 | 37.02/38.86/43.76 | 27.98/54.55/58.04 | 56.65/58.72/69.27 | 64.43/81.69/83.58 | 52.02/52.96/53.94 | 53.74/62.63/66.36 |
| | Random | 84.20/88.57/89.06 | 36.80/38.08/44.40 | 25.36/53.57/55.36 | 55.74/56.80/67.19 | 62.23/80.46/81.44 | 55.02/56.31/53.86 | 53.23/62.30/65.22 |
| | Cluster | 85.34/88.96/90.90 | 37.22/39.50/46.26 | 33.24/52.87/58.15 | 64.88/67.83/71.04 | 68.44/83.52/84.93 | 51.88/52.45/54.02 | 56.83/64.19/67.55 |
| | DPP | 86.32/89.67/90.81 | 38.40/40.60/53.40 | 47.47/56.10/62.07 | 74.26/76.30/80.54 | 75.03/86.28/88.95 | 52.65/54.90/63.18 | 62.36/67.31/73.16 |
| | Ours | 88.20/91.67/92.61 | 58.02/67.60/72.78 | 49.50/58.66/64.29 | 79.66/82.26/84.09 | 80.45/89.26/92.47 | 53.02/55.48/56.63 | 68.14/74.16/77.15 |
| GPT-NeoX 20B | LC | 82.55/85.47/89.14 | 42.18/51.53/60.89 | 46.99/58.74/60.02 | 69.43/72.29/75.86 | 70.08/78.24/83.04 | 54.82/55.67/57.88 | 61.01/66.99/71.14 |
| | Cal | 84.40/90.01/93.22 | 44.72/54.88/61.97 | 50.05/61.37/62.18 | 73.62/75.47/79.15 | 74.20/83.09/84.22 | 56.44/57.05/57.92 | 63.91/70.31/73.11 |
| | Random | 83.57/89.25/92.89 | 44.56/53.80/63.12 | 49.13/60.71/60.00 | 74.13/73.73/77.91 | 72.66/81.84/83.68 | 57.26/59.06/57.83 | 63.55/69.73/72.57 |
| | Cluster | 84.48/90.39/93.02 | 44.88/55.12/63.87 | 51.18/61.27/62.05 | 75.89/76.78/80.42 | 74.05/82.56/84.32 | 57.40/58.02/58.34 | 64.65/70.69/73.67 |
| | DPP | 86.59/93.19/93.41 | 46.00/57.20/65.40 | 50.00/62.93/63.29 | 77.36/80.34/83.43 | 80.76/89.32/91.08 | 58.48/56.48/58.87 | 66.53/73.24/75.91 |
| | Ours | 88.76/93.80/94.48 | 61.12/71.03/79.94 | 54.18/65.56/69.27 | 81.57/84.76/86.48 | 84.14/92.27/93.62 | 58.43/58.87/59.22 | 71.37/77.72/80.50 |

Table 1: 2-shot/4-shot/8-shot performance comparison on six datasets across three LLMs. AVG denotes the average performance of the whole six datasets.

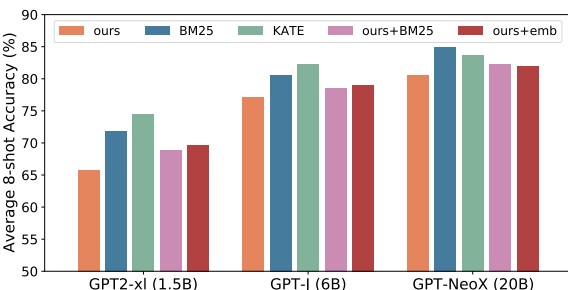

Figure 3: Comparsions of average 8-shot performance of the six datasets across the three LLMs.

achieve better performance compared to Cluster, which indicates that DPP is the better choice to introduce diversity. Compared to these baselines, our methods significantly improve the performances among all datasets in different settings across these three LLMs. In specific, compared to random, our methods bring more than 10% improvements across the three shots for GPT2-xl and GPT-J and 8% improvements across the three shots for GPT-NeoX. Compared to DPP (only stage 1, §3.2.1), which is the strongest baseline, our method also achieves an improvement ranging from 4% to 7% under different settings, which verifies the effectiveness of the second stage (§3.2.2).

### 4.3 Comparison with Retrieval Methods

**Performance Comparison** We compare our method to the retrieval-based methods BM25 (Wang et al., 2022) and KATE (Liu et al., 2022), which retrieve the demonstrations according to the BM25 scores (Liu et al., 2009) and the RoBERTa embeddings (Liu et al., 2019). Figure 3 shows the average 8-shot performance of the six datasets. From these results, it is not surprising that these two methods could achieve better performance. This is because they both introduce information about the test set while our method does not.

To make a fair comparison, we also try to incorporate test set information into our method. Specifically, we used a retriever to find the most similar examples from the training set to construct the score set $\mathcal{D}_{\text{score}}$ for our method. We name the variants using the BM25 retriever in BM25 and the Roberta retriever in Kate as Ours+BM25 and Ours+Emb, respectively. According to the results, these two variants achieve further performance improvements and the performance gaps with retrieval methods decrease as the model size increases.

**Benefits of Our Method** Actually, it is inevitable that the performance of representative demonstrations is inferior to that of retrieval methods, as they provide different demonstrations for each test instance. However, providing the same demonstration subset (our method) can save a significant amount of resources in real-world scenarios (Cheng et al., 2023). In specific, if the demonstration subset is kept the same, they can be pre-encoded[1], thus saving a significant amount of time spent on demonstration encoding when prediction. Figure 4 shows a comparison of the time spent on inferring 5000 instances on AGNews. With the support of pre-

---

[1]This paradigm has been widely implemented in Huggingface and these pre-encoded sentences are called past_key_values in current available LLMs.

|        | BM25  | KATE  | Ours+batch prompting |
|--------|-------|-------|----------------------|
| 2-shot | 15357 | 15349 | 4093                 |
| 4-shot | 26653 | 26677 | 4160                 |
| 8-shot | 49241 | 46274 | 4160                 |

Table 2: Comparisons of token costs when testing Trec with `text-davinci-002` API, whose maximum input length is 4097. We list the results of three shots.

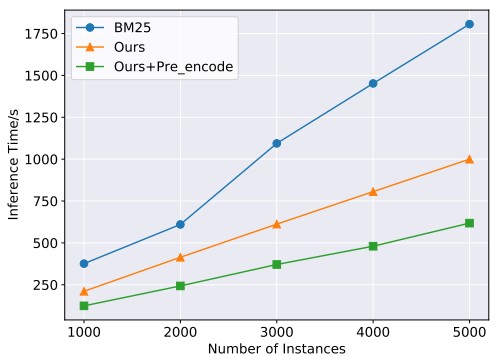

Figure 4: Comparsions of inference time of 5000 instances from AGNews for GPT-J.

encode, only about 1/3 of the inference time is required. In the API calling scenario, the cost is calculated based on tokens. Cheng et al. (2023) proposes batch prompting, which could generate responses for multiple examples in one batch using the same demonstration set. We calculate the token costs when testing Trec with `text-davinci-002`. Table 2 presents the comparisons. With batch prompting, ICL with the same demonstration subset shows a significant advantage in token usage, requiring only 1/10 of the token costs of the retrieval-based method for 8-shot learning.

Therefore, we believe that representative demonstration selection is the more practical paradigm, which could save a significant amount of inference time and token costs.

### 4.4 Effect of Factors: Semantic Diversity, Instance Quality, and Influence Diversity

In this part, we explore the effect of the proposed three key factors in our selection method: semantic diversity, instance quality, and influence diversity.

Table 3 presents the corresponding results and Appendix B shows the detail of each baseline. From these results, we find that instance quality is the most important factor, which contributes significantly to performance improvement. Semantic diversity also plays a significant role, which is consistent with the results of `Cluster` and `DPP`. And

| variant | SST-2 | Trec | AGNews |
|---------|-------|------|--------|
| random | 54.86 | 26.11 | 42.07 |
| sem_div | 57.58 | 32.20 | 66.82 |
| ins_qua | 61.93 | 37.08 | 68.11 |
| inf_div | 54.26 | 29.47 | 65.97 |
| sem_div + ins_qua | 63.86 | 39.92 | 69.88 |
| sem_div + inf_div | 59.29 | 34.87 | 68.08 |
| ins_qua + inf_div | 62.47 | 37.76 | 68.75 |
| sem_div + ins_qua + inf_div (ours) | 65.52 | 42.83 | 72.51 |

Table 3: Effect of the three factors: semantic diversity (`sem_div`), instance quality (`ins_qua`), and influence diversity (`inf_div`). We list the average performance among the three shots for GPT2-xl.

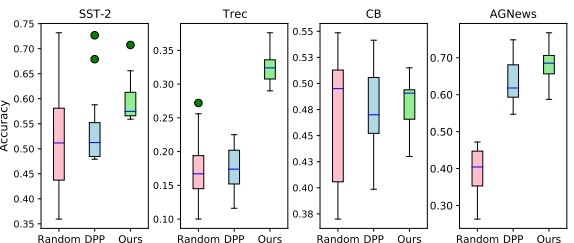

Figure 5: Order sensitivity on GPT2-xl. We conduct 4-shot experiments on SST2, Trec, CB, and AGNews and show the results of the total 24 orders.

the influence diversity would have a relatively positive effect when it appears with instance quality.

## 5 Discussions

### 5.1 Order Sensitivity of Representative In-Context Demonstrations

Lu et al. (2022) shows that ICL performance is highly sensitive to the demonstration order. Therefore, we also test whether the carefully selected representative demonstrations suffer from the order problem. In specific, we compare our method with random selection and the strongest baseline DPP.

Figure 5 presents the corresponding results on GPT2-xl. From these results, we could observe that DPP could reduce the order sensitivity of ICL and our method further reduce the order sensitivity. According to this, we conjecture that the selected representative in-context demonstrations could effectively characterize the corresponding task, leading to a reduction in order sensitivity of ICL, which is also consistent Chen et al. (2022).

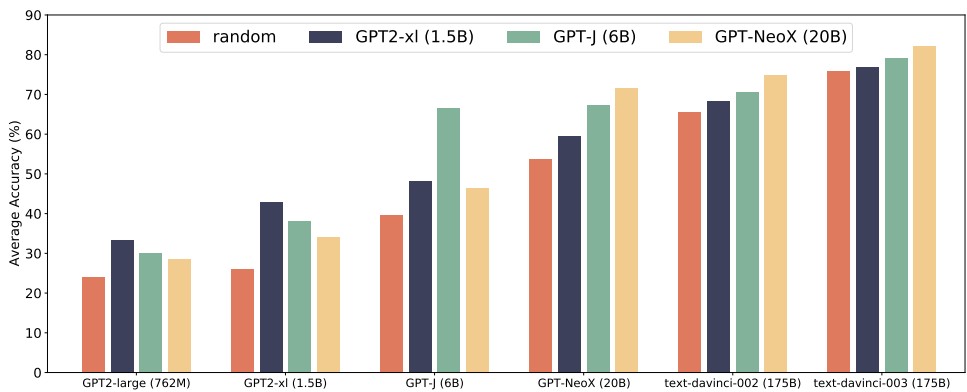

Figure 6: Transferability of representative demonstrations. We test the representative demonstrations selected based on GPT2-xl, GPT-J, and GPT-NeoX models on six different models GPT2-large, GPT2-xl, GPT-J, GPT-NeoX, text-davinci-002, and text-davinci-003. We show the average performance on Trec of 2-/4-/8- shots.

## 5.2 Transferability of Representative In-Context Demonstrations across LLMs

In this section, we investigate the transferability of the selected representative demonstrations.

In specific, we utilize the demonstrations selected based on the original model to test its performance on the new model. We choose GPT2-xl, GPT-J, and GPT-NeoX which were employed in previous experiments as the original models, and GPT2-large, GPT2-xl, GPT-J, GPT-NeoX, text-davinci-002, and text-davinci-003 as the new models, whose scale varies from 762M to 175B. Figure 6 shows the corresponding results.

From these results, we observe that the selected representative demonstrations all achieve better performance compared to random selection, which verifies their transferability. Besides, we also find that the transferability will be better when the size of the original model and the new model is closer.

## 5.3 Assessing the Quality of the Selected In-Context Demonstrations

In this part, we explore the quality of the selected in-context demonstrations. Considering that it is not feasible to enumerate all possible combinations in a training dataset, we conduct simulations under the following conditions.

Specifically, we sample 10 instances from SST-2, yielding C(10,4) = 210 possible combinations for 4-shot experiments. Then we test each of these 210 combinations on the test set and obtained the corresponding test accuracy for each combination. Finally, we examine the ranking of the combinations selected by our method among all possible combinations.

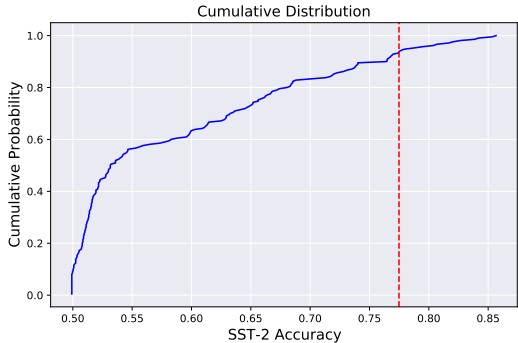

Figure 7: Cumulative distribution of the C(10,4) = 210 4-shot combinations on SST-2. And we sign our method with the red line, which outperforms more than 90% proportion of combinations.

As shown in Figure 7, we show the cumulative distribution of the 210 combinations. And our method is signed with the red line. From the results, we can find that our method outperforms 90% combinations, which reveals the effectiveness of our method.

## 5.4 Effect of Subet Size $\mathcal{D}_{\text{sem}}$

In Algorithm 1, we get a subset of size $\mathcal{D}_{\text{sem}}$ after the first stage DPP and select high-quality instances from this subset. However, it is inevitable to filter out some high-quality instances in the first stage. Therefore, we explore the effect of the subset size in this section. In our main experiments, we set $|\mathcal{D}_{\text{sem}}| = 200$, and we show the performances of more value here and show the results of SST-2 on GPT2-xl in Figure 8.

We could observe that our method achieves better performance with the bigger subset size across different shots. This indicates bigger subset size

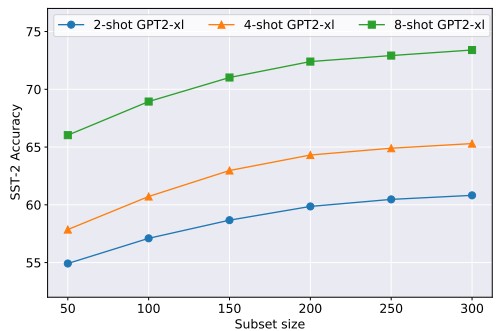

Figure 8: Effect of the subset size of SST-2 for GPT2-xl.

could introduce more high-quality instances in the candidate set. Based on this, the second stage could select better demonstrations. However, we could also observe that the performance improvements diminish significantly once the size exceeds 200. Considering that the bigger size would introduce much more inference costs when computing quality scores, this paper just selects 200 to conduct experiments for efficiency.

## 6 Related Work

### 6.1 In-Context Learning

Existing studies have shown ICL is sensitive to demonstration selection (Liu et al., 2022), demonstration formats (Dong et al., 2022), demonstration labels (Min et al., 2022; Yoo et al., 2022), and demonstration order (Lu et al., 2022). This paper focuses on the selection of demonstrations.

Existing methods aim to find different demonstrations for each test instance by retrieval-based methods. These methods can be divided into two categories based on whether the retriever requires training (Dong et al., 2022). The training-free methods utilize the sentence representations (Liu et al., 2022), BM25 (Wang et al., 2022), mutual information (Sorensen et al., 2022), and perplexity (Gonen et al., 2022) to select demonstrations. As for methods that require training, Zhang et al. (2022a) utilized reinforcement learning to train a retriever. Rubin et al. (2022) trained an example scorer using contrastive learning with signals from LLM. More recently, Li et al. (2023) integrated different tasks to train a universal retriever, further enhancing the performance of retrieval methods.

However, these methods have certain limitations in real-world applications, especially in more inference time and high token costs (§4.3). Based on this, this paper tries to solve a more challenging problem: selecting representative in-context

demonstrations, which could prompt each test example with the same demonstrations.

### 6.2 Determinantal Point Process

Determinantal Point Process (DPP) is an elegant probabilistic model that could select representative subsets while maintaining high diversity among each instance (Kulesza et al., 2012). Such efficient method has been applied to introduce diversity in various tasks: objection detection (Azadi et al., 2017), recommendation (Chen et al., 2018), summary (Cho et al., 2019), and parsing (Shi et al., 2021). More recently, Levy et al. (2023) using DPP in composition tasks, sampling a diverse subset of in-context examples to cover more sub-phrases, which is task-specific and dependent on the specific model. Ye et al. (2023) improve the retrieval-based methods by introducing diversity into the retrieved examples via DPP. However, this paper focuses on a totally different paradigm.

## 7 Conclusion

This paper aims to address the challenge of selecting a representative demonstration subset and proposes two criteria (*quality* and *diversity*) that such demonstrations should satisfy. And we further propose a two-stage DPP method to incorporate both high quality and diversity in the selection process. Extensive experimental results show the effectiveness of our method and the significant advantages compared to retrieval-based methods.

## 8 Ackonwledgements

This work was supported by the National Key R&D Program of China (2022ZD0160503), the National Natural Science Foundation of China (No.62276264 and No.62306087), and the Natural Science Foundation of Shandong Province (Grant No. ZR2023QF154). This research was also supported by Meituan. And we thank the reviewers for their valuable feedback.

## Limitations

The main limitation of this paper is the proposed method could not be transferred into the black-box scenario such as the GPT-3.5 family. Existing Openai APIs only provide the log probability of the top-5 tokens, which leads to inaccurate calculation of the influence score. With such an inaccurate influence score, the accuracy of the values of the

two important factors, instance quality and influence diversity, will also be affected. Therefore, our method is difficult to be transferred to these black-box models.

Besides, we have realized the defined influence score (Equation 1) is not the optimal choice. We also design a more accurate influence score and Equation 1 is just a special case. These more accurate scores would bring more performance improvements. However, the computational cost of this more accurate method is several tens of times higher than Equation 1 used in this paper. For large language models, this cost is unacceptable (Yang et al., 2023). Therefore, this paper made trade-offs between efficiency and performance. And we leave the challenge of how to efficiently incorporate the more accurate influence scores into our method as our future work.

## Ethics Statement

This paper aims to select representative demonstrations for in-context learning, and the experiments are conducted on publicly available datasets with available LLMs. As a result, there is no data privacy concern. Meanwhile, this paper does not involve human annotations, and there are no related ethical concerns.

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

## A    Details of Datasets

Table 4 shows the data statistics of our used datasets in the experiments. Considering the test set of DBPedia is too large, we just sample 3000 examples for testing in retrieval-based comparisons (Figure 3).

| Dataset | Class | Train | Test |
|---------|-------|-------|------|
| SST-2   | 2     | 6921  | 1821 |
| Trec    | 6     | 5452  | 500  |
| CB      | 3     | 250   | 250  |
| AGNews  | 4     | 120000| 7600 |
| DBPedia | 14    | 50000 | 70000|
| RTE     | 2     | 2490  | 3000 |

Table 4: Stastics of the datasets.

Table 5 shows the prompt formats for the six datasets. Following Zhao et al. (2021), we let the model predict the label for the given sentence for SST-2 and AGNews. For TREC and DBPedia, we add the instruction sentence to illustrate the label space. For CB and RTE, which is a sentence matching task, we take the first sentence as background and combine the second sentence with a question as a prompt.

## B    Details of the Ablation Baseline

Table 3 presents the effect of the three factors. In this part, we illustrate the details of each baseline.

sem_div introduces semantic diversity via DPP and chooses $k$ demonstrations, which is the same as the DPP baseline and similar to the first stage (§3.2.1). ins_qua randomly samples the candidate subset and computes the influence score of each instance. Then we choose the top-$k$ samples with the highest quality scores as demonstrations. ins_qua also randomly samples the candidate subset and computes the influence score of each instance. Then we introduce influence diversity via DPP and choose $k$ demonstrations. sem_div+ins_qua obtains the candidate subset by semantic DPP and computes the influence score of each instance. Then we choose the top-$k$ samples with the highest quality scores as demonstrations. sem_div+inf_div obtains the candidate subset by semantic DPP and computes the influence score of each instance. Then we introduce influence diversity via DPP and choose $k$ demonstrations. ins_qua+inf_div randomly samples the candidate subset and computes the influence score of

each instance. Following the second stage (§3.2.2), we utilize DPP to incorporate both quality and influence diveristy to choose $k$ demonstrations.

## C    Frame of Our Method

To provide a more direct illustration of our method, we present the overall workflow in Figure 9, which aids in better understanding Algorithm 1.

| Task | Prompt | Label Names |
|---|---|---|
| SST-2 | Review: [sentence]
Sentiment: | Positive, Negative |
| AGNews | Article: [sentence]
Answer: | World, Sports, Business, Technology |
| TREC | Classify the questions based on whether their answer type is a Number, Location, Person, Description, Entity, or Abbreviation.

Question: [sentence]
Answer Type: | Number, Location, Person, Description, Entity, Abbreviation |
| DBPedia | Classify the documents based on whether they are about a Company, School, Artist, Athlete, Politician, Transportation, Building, Nature, Village, Animal, Plant, Album, Film, or Book.

Article: [sentence]
Answer: | Company, School, Artist, Athlete, Politician, Transportation, Building, Nature, Village, Animal, Plant, Album, Film, Book |
| CB | [sentence1]
question: [sentence2] True, False, or Neither?
answer: | True, False, Neither |
| RTE | [sentence1]
question: [sentence2] True or False?
answer: | True, False |

Table 5: The prompt formats used for the six datasets.

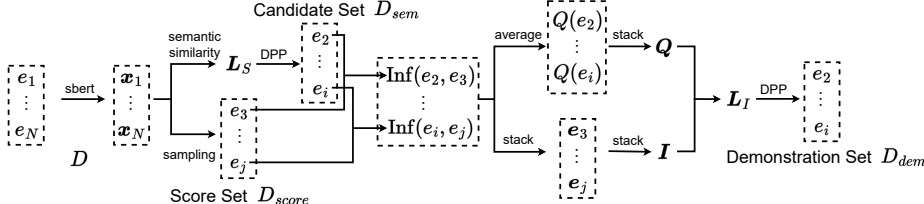

Figure 9: Total frame of our method (Algorithm 1).