# OpenReview forum: "Representative Demonstration Selection for In-Context Learning with Two-Stage Determinantal Point Process"
_EMNLP/2023/Conference — EMNLP 2023 Main_

### Official Review · Reviewer_twH8 · 2023-07-30

**Soundness:** 3

**Excitement:**

3: Ambivalent: It has merits (e.g., it reports state-of-the-art results, the idea is nice), but there are key weaknesses (e.g., it describes incremental work), and it can significantly benefit from another round of revision. However, I won't object to accepting it if my co-reviewers champion it.

**Paper Topic And Main Contributions:**

The research paper focuses on the concept of In-Context Learning (ICL) with Large Language Models (LLMs) and its limitations, particularly in relation to demonstration selection. ICL has shown significant improvements over zero/few-shot settings, yet, it is sensitive to factors like demonstration selection, format, labels, and order. The authors emphasize the importance of choosing suitable demonstrations.
To alleviate the problem of low efficiency of existing methods, the authors offer a two-stage solution to meet these criteria. The first stage involves using semantic information to measure instance similarity and leveraging a DPP to obtain a subset with high semantic diversity. The second stage involves computing the quality score of each instance in the subset based on influence scores. Then another DPP is constructed using these scores to incorporate quality and influence diversity in the selection process. This method aims to optimize the selection process, ensuring it covers more semantics in the training set and improves classification performance.

**Reasons To Accept:**

1)	The paper is easy to follow, and the logical structure of the article is clear. Furthermore, the authors propose two criteria (quality and diversity) that in-context demonstrations should satisfy.
2)	The experimental effects achieve great improvements, and the results show a significant advantage in inference time and token usage compared to the retrieval-based method.

**Reasons To Reject:**

1)	The baseline method compared in the article is somewhat outdated, currently in 2023, with only one baseline method in 2022.
2)	The author should open source code to enhance the persuasiveness of the article and increase its contribution to the research community.
3)	The author may consider adding a model frame diagram to better understand the entire process.
4)	Is it appropriate for the authors to use the DDP process to achieve semantic diversity, high quality, and influence diversity, as these features are considered in stages.

**Reproducibility:**

2: Would be hard pressed to reproduce the results. The contribution depends on data that are simply not available outside the author's institution or consortium; not enough details are provided.

**Reviewer Confidence:**

2: Willing to defend my evaluation, but it is fairly likely that I missed some details, didn't understand some central points, or can't be sure about the novelty of the work.

---

> ### Author Rebuttal · Authors · 2023-08-28
>
> Thank you for your time and constructive review! We address your questions and concerns below:
>
> 1. Weakness 1: The baseline method compared in the article is somewhat outdated, currently in 2023, with only one baseline method in 2022.
> * Reply: Thanks for your suggestion. This paper aims to find representative demonstrations for the whole testing dataset in ICL, where this setting seems to be the firstly addressed. More recently, we indeed found some newer baselines that are also suitable for this problem ([1]). In summary, their methods focus on semantic diversity, sharing similar motivations with the cluster and DPP baselines in our paper. Considering their core idea focuses on only semantic diversity, we believe our method could get better performance.  We will add the comparisons with these newer baselines in the next version.
>
>   [1] What Makes Good In-context Demonstrations for Code Intelligence Tasks with LLMs? ASE2023
>
> 2. Weakness 2: The author should open source code to enhance the persuasiveness of the article and increase its contribution to the research community.
> * Reply: Thanks for your advice. We promise to release our source code in github.
>
> 3. Weakness 3: The author may consider adding a model frame diagram to better understand the entire process.
> * Reply: We will add a frame diagram to describe the entire process of our method (Algorithm 1) in the next version. We believe that it could indeed make our paper easier to understand. Thanks for your valuable suggestion.
>
> 4. Weakness 4: Is it appropriate for the authors to use the DDP process to achieve semantic diversity, high quality, and influence diversity, as these features are considered in stages.
> * Reply: We admit that if we incorporate these three factors into a single DPP process would solve your concerns. However, the latter two factors, high quality and influence diversity both require the computation of the quality score of each instance. It is infeasible to compute the quality score for each instance in the total training set because of the computation costs. Therefore, this paper employs a two-stage method and we believe the two-stage method could also incorporate these three factors. After the first stage DPP, the obtained demonstration set satisfies the semantic diversity. With this set, we could compute the quality score for each instance and incorporate high quality and influence diversity into the second stage DPP. Therefore, we think our two-stage method could also incorporate these three factors into the selection process.

---

### Official Review · Reviewer_4SPy · 2023-08-03

**Soundness:** 4

**Excitement:**

4: Strong: This paper deepens the understanding of some phenomenon or lowers the barriers to an existing research direction.

**Missing References:**

Xiaonan Li, Kai Lv, Hang Yan, Tianyang Lin, Wei Zhu, Yuan Ni, Guotong Xie, Xiaoling Wang, Xipeng Qiu. Unified Demonstration Retriever for In-Context Learning. ACL 2023

**Paper Topic And Main Contributions:**

The paper focuses on the key problem of selecting a representative subset of in-context demonstrations. It emphasizes two criteria for demonstration selection: quality and diversity. To achieve this, the paper introduces a two-stage Determinantal Point Process (DPP) method, aiming to find a representative subset that bolsters model performance. Comprehensive experiments affirm the effectiveness of this method, presenting a robust comparison to retrieval-based methods.


**Questions For The Authors:**

* Could you clarify the meaning of the gray background in Table 1? It appears to denote the best result, but in the case of RTE 2-shot under GPT-NeoX 20B, the paper's method seems to perform worse than DPP. How is this reconciled?

**Reasons To Accept:**

* The paper addresses a crucial and timely issue in the field of large language models, making it highly relevant.
* The paper is well-written, clear, and easy to follow. The persuasive analysis in section 2, for instance, enhances the overall comprehensibility and presents strong arguments supporting the proposed method.
* The proposed two-stage DPP method is simple and effective. The solid experimental design and execution validate the method's efficacy.

**Reasons To Reject:**

* Although the paper mentions some related work, such as retrieval-based methods, it misses the opportunity to discuss recent similar works like [1]. Including such references would enrich the context and comparison.
* The experiments could benefit from incorporating newer baselines such as (Zhang et al., 2022a). Including more contemporary comparisons would strengthen the credibility and persuasiveness of the results.

[1] Xiaonan Li, Kai Lv, Hang Yan, Tianyang Lin, Wei Zhu, Yuan Ni, Guotong Xie, Xiaoling Wang, Xipeng Qiu. Unified Demonstration Retriever for In-Context Learning. ACL 2023

**Reproducibility:**

3: Could reproduce the results with some difficulty. The settings of parameters are underspecified or subjectively determined; the training/evaluation data are not widely available.

**Reviewer Confidence:**

4: Quite sure. I tried to check the important points carefully. It's unlikely, though conceivable, that I missed something that should affect my ratings.

---

> ### Author Rebuttal · Authors · 2023-08-28
>
> Thank you for your time and constructive review! We address your questions and concerns below:
>
> 1. Weakness 1: Although the paper mentions some related work, such as retrieval-based methods, it misses the opportunity to discuss recent similar works like [1]. Including such references would enrich the context and comparison.
> * Reply: Thanks for your advice. We would add the discussion about more recent retrieval-based methods including [1] in our next version to make our paper more complete.
>
>   [1]  Unified Demonstration Retriever for In-Context Learning. ACL 2023
>
> 2. Weakness 2: The experiments could benefit from incorporating newer baselines such as (Zhang et al., 2022a). Including more contemporary comparisons would strengthen the credibility and persuasiveness of the results.
> * Reply: Thanks for your valuable suggestion. Due to our oversight, we indeed failed to compare our method with the newer baseline ([2]) in 2023. Actually, this method primarily focuses on semantic diversity, sharing similar motivations with the cluster baseline in our paper. In the next version, we will add experimental comparison results of these updated baseline methods to enhance the persuasiveness of our paper.
>
>   [2] What Makes Good In-context Demonstrations for Code Intelligence Tasks with LLMs? ASE2023
>
> 3. Question 1: Could you clarify the meaning of the gray background in Table 1? It appears to denote the best result, but in the case of RTE 2-shot under GPT-NeoX 20B, the paper's method seems to perform worse than DPP. How is this reconciled?
> * Reply: We added a gray background just to highlight the results of our method in Table 1. We will add bold numbers to denote the best result and make the table clearer in our next version. In the case of 2-shot RTE under GPT-NeoX 20B, we review the results and find that DPP achieves much higher results (may be an outlier) in seed 2, which leads to the average performance even surpassing our method.

---

### Official Review · Reviewer_3tF3 · 2023-08-04

**Soundness:** 4

**Excitement:**

3: Ambivalent: It has merits (e.g., it reports state-of-the-art results, the idea is nice), but there are key weaknesses (e.g., it describes incremental work), and it can significantly benefit from another round of revision. However, I won't object to accepting it if my co-reviewers champion it.

**Paper Topic And Main Contributions:**

The paper aims to select a representative demonstration subset for a task in ICL, to satisfy high quality and diversity at the same time. The two criteria are explicitly defined in the paper, followed by the empirical analyses, demonstrating that meet these criteria can indeed bolster model performance. To achieve the goal, a method based on two-stage Determinantal Point Process (DPP) is proposed. Experimental results show the effectiveness of the method.

**Reasons To Accept:**

1. Influence diversity is proposed which could support the paradigm sharing the same subset.
2. Pilot experiments are given to explore whether the proposed criteria could improve the performance of ICL.
3. The method can obviously reduce the time-costs in the selection stage for each testing example.

**Reasons To Reject:**

1. Intuitively, the "diversity" may average or smooth the distribution of the selected samples.
Q1: How can we guarantee that the useful demonstration will not be filtered out?
Q2: How can we guarantee that the obtained subset is the optimal distribution in terms of the whole dataset?
2. The proportion of the selected subset should be explored, and for the unselected samples it needs to further investigate that whether there are useful samples that are dropped falsely.
3. The motivation of Representative-based method, compared with Retrieval-based approaches, is not very intuitive, since all the testing samples share the same subset although "influence diversity" could somewhat solve the issue.

**Reproducibility:**

3: Could reproduce the results with some difficulty. The settings of parameters are underspecified or subjectively determined; the training/evaluation data are not widely available.

**Reviewer Confidence:**

3: Pretty sure, but there's a chance I missed something. Although I have a good feel for this area in general, I did not carefully check the paper's details, e.g., the math, experimental design, or novelty.

---

> ### Author Rebuttal · Authors · 2023-08-28
>
> Thank you for your time and constructive review! We address your questions and concerns below:
>
> 1. Weakness 1: Intuitively, the "diversity" may average or smooth the distribution of the selected samples. Q1: How can we guarantee that the useful demonstration will not be filtered out? Q2: How can we guarantee that the obtained subset is the optimal distribution in terms of the whole dataset?
> * Reply For Q1: We agree that some useful demonstrations may be filtered out in the first stage. The first stage only employs semantic diversity to select a candidate subset and it is inevitable to filter out some useful demonstrations in this process. However, in our experiments, we also observe that this will have minimal effects on the final performance. In specific, we conducted an experiment on the CB dataset, whose training set is small. Therefore, we can exclude the first stage that introduces semantic diversity and instead compute quality scores for all training instances. However, the results indicated a minimal improvement of less than 1%.
>
>   Actually, for k-shot ICL, we just need k demonstrations and the number of useful demonstrations may be far more than k for a big training set such as AGNews (12000 training instances). Therefore, it is inevitable to filter out some useful demonstrations in the first stage. Besides, the second stage ensures that the selected k demonstrations are all of high quality. And the influence diversity factor further ensures that the composition of these demonstrations would support more test instances. Therefore, incorporating more useful demonstrations may not necessarily lead to a significant performance improvement. Another thing to consider is that the cost of determining the usefulness of each example in the entire training set is enormous. Therefore, this paper just takes a trade-off and introduces the influence diversity factor to alleviate the problem. And we believe this would be a crucial problem for representative demonstration selection and we would take this as our future work. Thanks for your valuable suggestion.
>
> * Reply For Q2: As for the second problem, the obtained subset indeed may be not the optimal distribution of the whole dataset. Actually, this problem could be seen as coreset selection problem, which has been explored in the past two years in machine learning and deep learning research areas [1]. And coreset selection has been regarded as NP-hard problem [2]. In this paper, we also take classical coreset selection methods Least Confidence and Cal as baselines. The experimental results indicate that these effective methods in deep learning do not perform well in ICL. For comparison, DPP could achieve good performance. Based on these empirical results, we think DPP could help obtain a good distribution of the whole dataset for ICL.  Furthermore, our method employs DPP to incorporate the three factors into the selection process.
>
>   To further alleviate your concerns, we will add the following experiments in the next version: Suppose the training set has only 10 instances, there are C(10,4)=210 4-shot subsets. Then we can inference 210 times to obtain the test performance of each 4-shot subset. In this way, we can know the position of the subset obtained from our method among all 210 subsets. We will add these experimental results in the next version to make our paper more complete. And we apologize that we cannot show more theoretical support because the mechanism of ICL is still unexplored now. And we believe that how to obtain optimal subset distribution of the whole dataset for ICL is indeed a problem worth long-term research like coreset selection in deep learning and machine learning.
>
>   [1] DeepCore: A Comprehensive Library for Coreset Selection in Deep Learning, DEXA2022
>
>   [2] Coresets for data-efficient training of machine learning models, ICML2020
>
> 2. Weakness 2: The proportion of the selected subset should be explored, and for the unselected samples it needs to further investigate that whether there are useful samples that are dropped falsely.
> * Reply: Thanks for your valuable suggestion. We agree that the proportion of the selected subset should be explored. In specific, we conducted experiments under GPT2-xl and tested the subset size of 50, 100, 150, 200, 250, and 300 on SST-2. The performance increases with bigger size but the improvements diminish significantly once the size exceeds 200. Considering that the bigger size would introduce much more inference costs when computing quality scores, this paper takes 200 for the experiments. We would add the effects of the subset size in the next version to make our paper more complete.
>
>  | Subset Size |  50 | 100 | 150 | 200 | 250 | 300 |
>  | :----: | :----: | :----: | :----: | :----: | :----: | :----: |
>  | 2-shot SST-2| 54.92 | 57.10 | 58.67 | 59.86 | 60.47 | 60.82 |
>  | 4-shot SST-2| 57.86 | 60.72 | 62.97 | 64.31 | 64.90 | 65.30 |
>  | 8-shot SST-2| 66.03 | 68.94 | 71.02 | 72.40 | 72.92 | 73.40 |
>
>
>
> 3. Weakness 3: The motivation of Representative-based method, compared with Retrieval-based approaches, is not very intuitive, since all the testing samples share the same subset although "influence diversity" could somewhat solve the issue.
> * Reply: The motivation of comparing Representative-based method with Retrieval-based approaches is to show the inference efficiency in our setting. Selecting the same demonstration set in Representative-based method would decrease inference time and API costs compared with Retrieval-based approaches that select different demonstrations for each testing instance (Line 446-474). Figure 4 shows that Representative-based method only requires 1/3 inference time and Table 2 shows that Representative-based method only requires 1/10 API costs, which is more applicable. And the performance of Retrieval-based methods could be intuitively regarded as the upper bound of Representative-based method and we also try to narrow this gap (Figure 3).

---

### Meta-Review · Area_Chair_VmCN · 2023-09-13

**Recommendation:** 4

**Metareview:**

**Strengths**:

1. Critical problem of selecting a representative subset of in-context demonstrations that can effectively prompt different test instances in a speciﬁc task.

2. high quality and diversity are stressed together for good quality examples.

3. two-stage Determinantal Point Process (DPP) method is simple and effective. results show a significant advantage in inference time and token usage compared to the retrieval-based method.


**Weaknesses**:


1. The proportion of the selected subset should be explored, and for the unselected samples it needs to further investigate that whether there are useful samples that are dropped falsely. Although the authors show results for GPT2-XL in rebuttal, they should include such analysis for other models also.

2. Missing latest baselines: comparisons with Xiaonan Li et al, ACL 2023; Zhang et al., 2022a.

3. Code is not made public.

**Suggestions**:


1. C(10,4)=210 4-shot subset quality evaluation and checking rank of selected subset should be done at least for a few samples.

2. Good to add a frame diagram to describe the entire process of our method (Algorithm 1).

3. Please address above weaknesses also in the revised draft.

---

### Decision · Program_Chairs · 2023-10-07

**Decision:**

Accept-Main

**Comment:**

**Strengths**:

1. Critical problem of selecting a representative subset of in-context demonstrations that can effectively prompt different test instances in a speciﬁc task.

2. high quality and diversity are stressed together for good quality examples.

3. two-stage Determinantal Point Process (DPP) method is simple and effective. results show a significant advantage in inference time and token usage compared to the retrieval-based method.


**Weaknesses**:


1. The proportion of the selected subset should be explored, and for the unselected samples it needs to further investigate that whether there are useful samples that are dropped falsely. Although the authors show results for GPT2-XL in rebuttal, they should include such analysis for other models also.

2. Missing latest baselines: comparisons with Xiaonan Li et al, ACL 2023; Zhang et al., 2022a.

3. Code is not made public.

**Suggestions**:


1. C(10,4)=210 4-shot subset quality evaluation and checking rank of selected subset should be done at least for a few samples.

2. Good to add a frame diagram to describe the entire process of our method (Algorithm 1).

3. Please address above weaknesses also in the revised draft.